

# Massive barnacle recruitment on the Gulf of St. Lawrence coast of Nova Scotia (Canada) in 2024 linked to increased sea surface temperature

Ricardo A. Scrosati[1] and Julius A. Ellrich[2]

[1] Department of Biology, St. Francis Xavier University, Antigonish, Nova Scotia, Canada
[2] Biologische Anstalt Helgoland, Alfred Wegener Institute Helmholtz Centre for Polar and Marine Research, Helgoland, Germany

## ABSTRACT

With the ongoing climate and oceanographic change, an increasing number of studies are reporting dramatic population losses caused by thermal extremes in intertidal habitats. Under moderate warming, however, populations can fare better in places where species normally experienced suboptimal temperatures. This article reports the massive recruitment of the barnacle *Semibalanus balanoides* on the Gulf of St. Lawrence coast of Nova Scotia (Canada) in 2024. As recruits appear mostly during May in this region, coastal sea surface temperature (SST) in April is critical for the ecological performance of larvae, as they are pelagic and live in the water column for weeks before intertidal settlement. Thus, a study that spanned 12 years (2005 to 2016) on this coast found that annual barnacle recruitment was positively correlated to April SST. In April 2024, coastal SST was 116% higher than for the same month averaged over those 12 years (4.1 *vs.* 1.9 °C). This SST spike was followed by an elevated recruitment that was 111% higher than the average for those 12 years (1,278 *vs.* 607 recruits dm$^{-2}$). Overall for the studied years, the amount of variation in annual barnacle recruitment statistically explained by April SST was 51%. While the southern distribution limit of *S. balanoides* has moved northwards in recent decades due to lethal warming, our results support the notion of improving reproductive success with seawater warming on colder northern shores.

## INTRODUCTION

The ongoing climate and oceanographic change is altering species abundance in many ecosystems (*Parmesan, 2006*; *Blowes et al., 2019*; *Wernberg et al., 2024*). Rocky intertidal systems occur between the lowest and highest tide marks on marine rocky shores and are no exception. As for all other ecosystems, temperature is a major factor affecting the abundance of rocky intertidal species. The changing thermal regimes that characterize the current climate and oceanographic change are affecting intertidal organisms in unusual ways (*Meunier, Hacker & Menge, 2024*). Many reported effects are negative, often involving

Corresponding author
Ricardo A. Scrosati, rscrosat@stfx.ca

drastic reductions in population size after thermal extremes (*Thomsen et al., 2021*; *Raymond et al., 2022*; *Cameron & Scrosati, 2023*). Under less extreme conditions, however, population changes can still occur but may not be necessarily negative. For instance, if warming at a given location exposes a species to values closer to its thermal optimum, local abundance may ultimately increase. These responses are implicit in recent poleward changes in the distribution of many species especially concerning their leading edge (*Chen et al., 2011*; *Poloczanska et al., 2013*; *Mieszkowska et al., 2021*).

To detect unusual population changes, field monitoring should produce data that can be compared with data for several previous years to get an idea of natural variation. In this study, we follow this principle to document a marked increase in intertidal barnacle recruitment on the Gulf of St. Lawrence coast of Nova Scotia (Canada) with the warmer waters registered in 2024. Barnacles are sessile organisms that are common in rocky intertidal habitats around the world. These organisms serve as food for upper trophic levels and can act as foundation species due to the habitat complexity they provide with their shells (*Anderson, 1994*; *Harley, 2006*). Barnacle larvae are pelagic, so recruitment for barnacles refers to the appearance of new organisms in a benthic habitat following the settlement and metamorphosis of larvae.

Barnacle recruitment is often studied not only due to the ecological significance of these organisms but also because their benthic recruitment is an indicator of nearshore pelagic conditions (*Navarrete et al., 2005*; *Scrosati & Ellrich, 2018*; *Lathlean et al., 2019*; *Valqui et al., 2021*; *Román et al., 2022*). The Gulf of St. Lawrence is one of the southernmost large bodies of water in the northern hemisphere that freezes in winter (*Saucier et al., 2003*). As a result, ice scour is a significant driver of intertidal community structure (*Scrosati & Heaven, 2007*). After the sea ice melts at the end of the winter, the rocky intertidal substrate appears mostly bare in wave-exposed habitats, as they are subjected to intense ice scour in winter. This substrate availability allows for active barnacle recruitment in the spring (*MacPherson & Scrosati, 2008*). Field surveys done at a representative location of the Gulf of St. Lawrence coast of Nova Scotia for 12 years from 2005 to 2016 measured annual barnacle recruitment at mid-to-high elevations in wave-exposed habitats, producing a reference dataset for the future (*Scrosati & Ellrich, 2016*). In Nova Scotia, there is only one intertidal species of barnacle (*Semibalanus balanoides*) and it undergoes recruitment in May and June, although most recruits appear in May (*Scrosati & Holt, 2021*). Since their nauplius larvae are in the water for 5 to 6 weeks (*Bousfield, 1954*; *Drouin, Bourget & Tremblay, 2002*), pelagic conditions in April are critical for larval growth and survival and, ultimately, benthic recruitment. In fact, an analysis of coastal sea surface temperature (SST) revealed that April SST was the best explanatory variable for barnacle recruitment, showing a positive relationship for the 12 years surveyed between 2005 and 2016 (*Scrosati & Ellrich, 2016*). As nauplius larvae feed on phytoplankton (*Anderson, 1994*; *Leal et al., 2024*), data on the concentration of chlorophyll-a in coastal waters (Chl-a, used as a proxy for phytoplankton abundance) were also analyzed and revealed that April Chl-a had a positive statistical influence on barnacle recruitment but smaller than April SST (*Scrosati & Ellrich, 2016*).

The present study was motivated by the massive barnacle recruitment detected in 2024, which broke all previous recruitment records for this coast. We hereby provide comparisons of barnacle recruitment in 2024 with the 2005–2016 period and analyze whether April SST and Chl-a were related to this unforeseen recruitment spike.

## MATERIALS AND METHODS

The baseline reference data (2005–2016) on annual barnacle recruitment were measured in June of those 12 years at Sea Spray (45°46.38′N, 62°8.67′W; Fig. 1), which is a representative location of the Gulf of St. Lawrence coast of Nova Scotia, Canada (*Scrosati & Ellrich, 2016*). The sampling design used in that study is summarized in this paragraph. The surveyed habitats are wave-exposed because they face open waters without any obstructions and are composed of stable volcanic bedrock. The intertidal range is 1.8 m on this coast. Each year, barnacle recruitment was measured in a single day of June (see dates in *Scrosati & Ellrich, 2024a*) along a permanent transect line located at 2/3 of this vertical range, that is, at 1.2 m of elevation (mid-to-high) above chart datum (lowest normal tide in Canada, representing 0 m of elevation). Barnacle recruitment was quantified as the density of recruits found in 29–33 (depending on the year) replicate quadrats (10 cm x 10 cm) positioned at random along the transect line. Because of intense ice scour every winter, barnacle recruits are by far the only macroscopic organisms in these habitats in June. Satellite data of coastal SST and Chl-a were retrieved from the Ocean Color website of the National Aeronautics and Space Administration (NASA). More details about the sampling design used for the 2005–2016 surveys are available in the article published by *Scrosati & Ellrich (2016)*.

In 2024, we measured the density of barnacle recruits in the same way. We counted all of the barnacle recruits found in 31 quadrats (10 cm × 10 cm) placed at random along the permanent transect line on 5 July. In Nova Scotia, more than 90% of intertidal barnacle recruitment takes place in May and recruit mortality by early July is negligible to null (*Scrosati & Holt, 2021*). Thus, our 2024 recruitment data were comparable to the 2005–2016 data. As for the previous years, we obtained SST and Chl-a data for April 2024 from NASA's Ocean Color website for the 4-km-×-4-km cell containing Sea Spray shore (*NASA, 2024*). We analyzed the data by conducting Welch and Pearson correlation tests, as appropriate, using Statistica v. 14. Since Kolmogorov–Smirnov tests revealed normality of the data, Levene's tests generally found homoscedasticity, and the obtained $P$ values were very low (see Results), no data transformations were necessary. The full dataset on barnacle recruitment is freely available from the figshare online repository (*Scrosati & Ellrich, 2024a*).

## RESULTS

In 2024, intertidal barnacle recruitment on the Sea Spray shore was extremely high, as recruits covered almost all available substrate at the time of our sampling (Fig. 2). Mean recruit density was $1,278 \pm 28$ recruits $dm^{-2}$ (mean ± SE) in 2024, the range for individual quadrats being 931–1,547 recruits $dm^{-2}$. In 2024, recruit density was significantly higher than for each of the 12 years surveyed between 2005 and 2016 ($P < 10^{-6}$ for each Welch

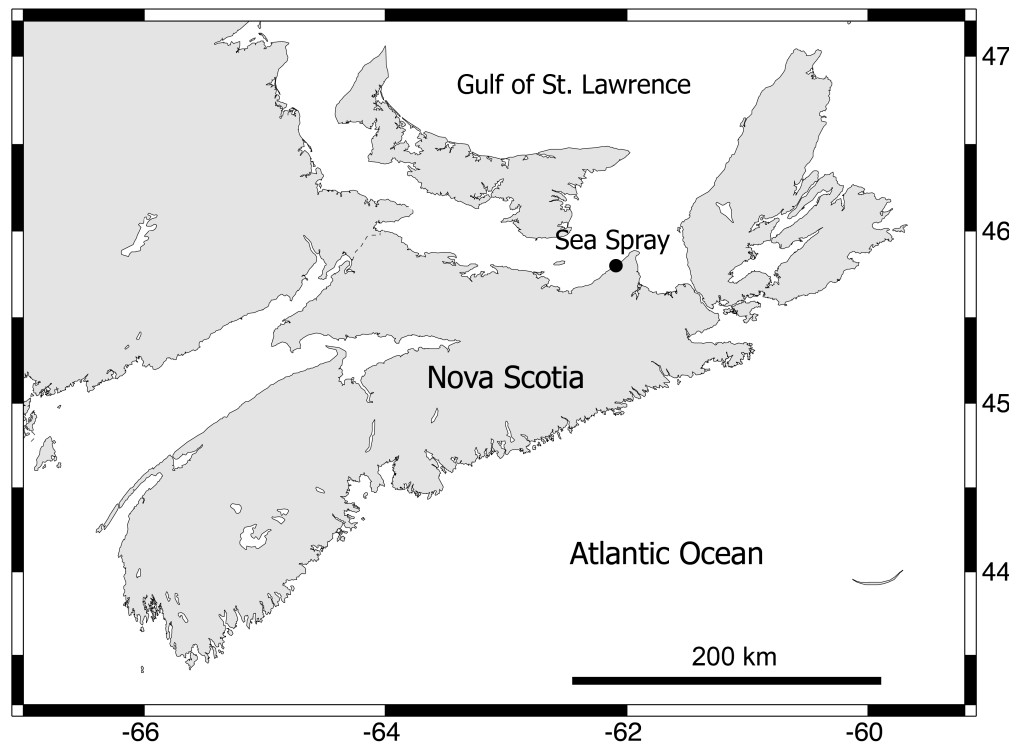

**Figure 1  Map of Nova Scotia.** Map indicating the studied location on the Gulf of St. Lawrence coast of Nova Scotia, Canada. Map source credit: *Scrosati & Ellrich (2024b)*, which is an open-access article whose copyright we own (DOI 10.7717/peerj.17697).

test), when mean annual recruitment ranged between 198 and 969 recruits $dm^{-2}$ (Fig. 3; see also *Scrosati & Ellrich, 2016*). In 2024, April SST was 4.1 °C, higher than for the month of April from 2005 to 2016, when monthly SST ranged between 0.6 and 3.1 °C (Fig. 3; Table 1). As a result, the correlation between annual barnacle recruitment and April SST remained significantly positive ($r = 0.71$, $P < 0.006$; Fig. 3). Combining the studied years (2005–2016 and 2024), April SST statistically explained more than half of the variation in annual barnacle recruitment (adjusted $R^2 = 51\%$). In 2024, however, April Chl-a (8.3 mg $m^{-3}$) was within the range measured in that month from 2005 to 2016 (1.4–12.6 mg $m^{-3}$; Table 1).

## DISCUSSION

On the Gulf of St. Lawrence coast of Nova Scotia, barnacle recruitment in wave-exposed habitats at mid-to-high intertidal elevations in 2024 was unusually high. The study done over 12 years between 2005 and 2016 had shown that barnacle recruitment did exhibit interannual changes and that coastal April SST explained 32% of that variation (*Scrosati & Ellrich, 2016*). However, in 2024, April SST was 116% higher than the April average for those 12 years (4.1 *vs.* 1.9 °C) and annual barnacle recruitment followed that trend by being 111% higher than the average for those 12 years (1,278 *vs.* 607 recruits $dm^{-2}$). Inclusion

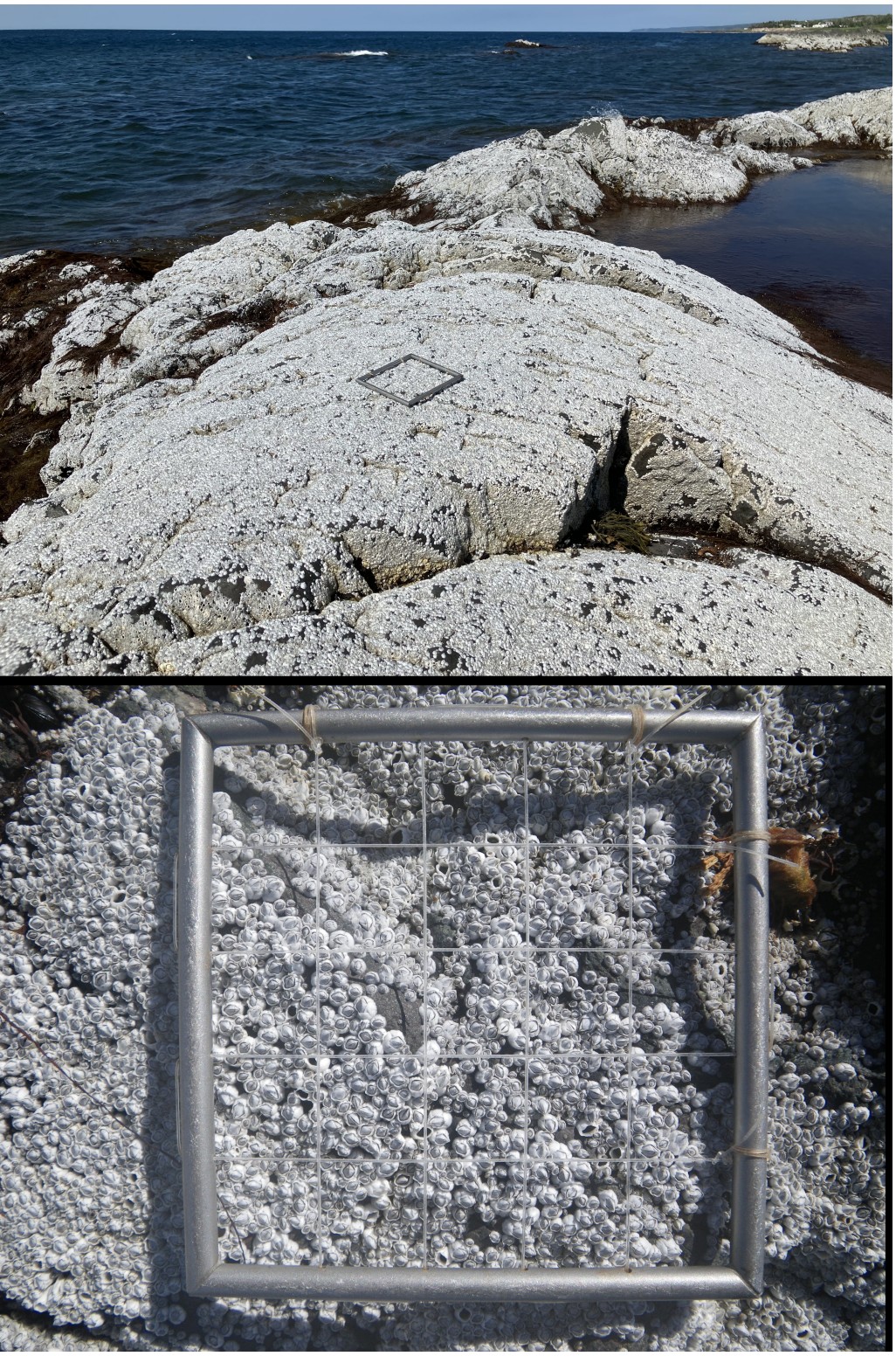

**Figure 2** **Barnacle recruitment on the Gulf of St. Lawrence coast of Nova Scotia in 2024.** Intertidal zone blanketed by barnacle recruits seen at low tide on 5 July 2024 (upper panel) and close-up view of barnacle recruits shown with a 10-cm-×-10-cm quadrat as a reference (lower panel). Photo credits: Ricardo A. Scrosati.

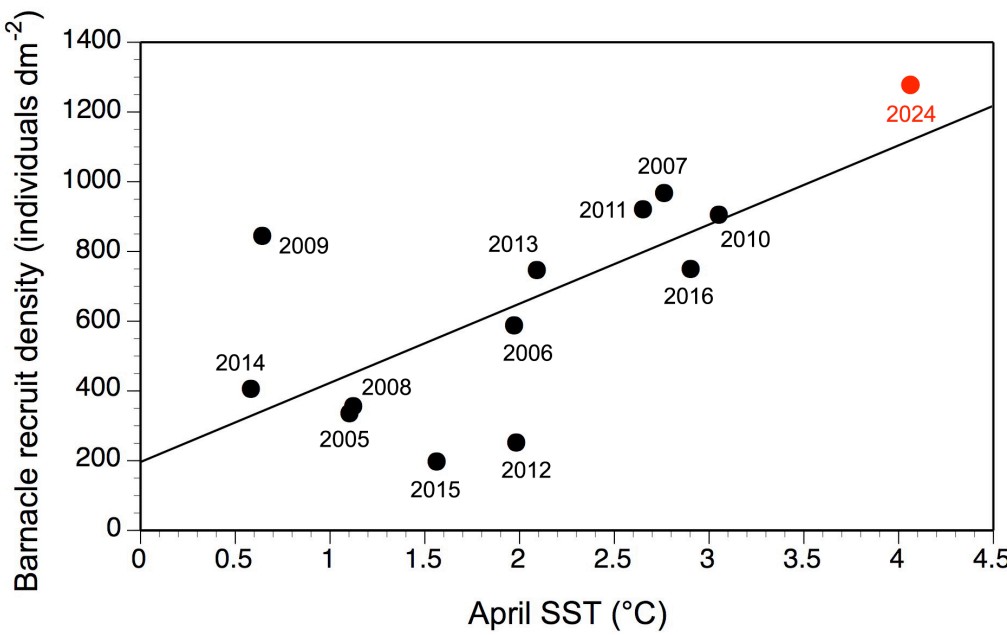

**Figure 3 Barnacle recruitment *versus* April SST.** Annual intertidal barnacle recruitment (expressed as recruit density) as a function of mean April SST.

**Table 1 SST and Chl-a.** April means of SST and Chl-a for 2005–2016 and 2024.

| Year | April SST (°C) | April Chl-a (mg/m$^3$) |
| --- | --- | --- |
| 2005 | 1.1 | 4.88 |
| 2006 | 1.97 | 6.03 |
| 2007 | 2.76 | 6.89 |
| 2008 | 1.12 | 6.49 |
| 2009 | 0.64 | 11.43 |
| 2010 | 3.05 | 7.76 |
| 2011 | 2.65 | 6.85 |
| 2012 | 1.98 | 9.79 |
| 2013 | 2.09 | 10.61 |
| 2014 | 0.58 | 12.63 |
| 2015 | 1.56 | 1.39 |
| 2016 | 2.9 | 5.67 |
| 2024 | 4.06 | 8.3 |

of 2024 thus raised the amount of variation in annual barnacle recruitment that could be explained by April SST (51%).

In Atlantic Canada, April SST is critical for the growth and survival of the pelagic larvae of *Semibalanus balanoides* and ultimately its annual recruitment, given that nauplius larvae stay in the water for 5–6 weeks (*Bousfield, 1954*; *Drouin, Bourget & Tremblay, 2002*) and that most recruitment occurs in May (*Scrosati & Holt, 2021*). Therefore, the massive

recruitment of 2024 may have resulted from improved larval performance as a result of warmer waters. Progressively warmer conditions caused by the ongoing global warming may thus increase barnacle recruitment on this coast in the future. Naturally, however, such increases may cease once the thermal optimum for this species is reached, which in turn may vary regionally. In Europe, for instance, the reproductive success of *S. balanoides* decreases beyond 7–17 °C, the threshold being lower for higher latitudes (*Abernot-Le Gac et al., 2013*; *Rognstad & Hilbish, 2014*; *Rognstad, Wethey & Hilbish, 2014*; *Herrera et al., 2021*). In fact, the southern distribution limit of this species has been moving northwards on both sides of the North Atlantic in recent decades as a consequence of lethal warming (*Crickenberger & Wethey, 2018*). Since the waters of the Gulf of St. Lawrence are comparatively cold in April (see Results), the reproductive success of *S. balanoides* might experience gains in the next several years given moderate levels of warming.

In contrast to SST, our data suggest that food supply for barnacle larvae did not play a role in the spike of barnacle recruitment seen in 2024. Between 2005 and 2016, April Chl-a had a small statistical effect on recruitment (*Scrosati & Ellrich, 2016*), but Chl-a in April 2024 did not increase as recruitment and SST did. As both SST and Chl-a are important for larval performance and benthic recruitment, their relative spatiotemporal variation seems to determine their respective influence. For example, on the open Atlantic coast of Nova Scotia (outside of the Gulf of St. Lawrence), Chl-a varied more along the coast than SST in April 2014 and, consequently, Chl-a had a larger statistical influence on barnacle recruitment than SST in that year (*Scrosati & Ellrich, 2018*).

The ongoing climate and oceanographic change is directly or indirectly modifying patterns of species distribution across the globe (*Parmesan, 2006*; *Chen et al., 2011*; *Poloczanska et al., 2013*; *Blowes et al., 2019*; *Mieszkowska et al., 2021*; *Wernberg et al., 2024*). Intertidal communities from Nova Scotia were not showing evident responses for years, but recently this situation has started to change. For example, an unusually severe cold spell decimated intertidal mussel populations in southeastern Nova Scotia in the winter of 2023 (*Cameron & Scrosati, 2023*), which was followed by the loss of additional mussel stands in the summer of that year after an unusually strong cyclone (*Scrosati, 2023*). Mussel recolonization, however, is taking place as of July 2024 (*Scrosati & Cameron, 2024*). The high barnacle recruitment associated to seawater warming herein reported for 2024 might be another example of these accelerating changes. The community-wide effects of a sustained increase in barnacle recruitment on the Gulf of St. Lawrence coast if water temperature continues to rise are unknown, but separate studies offer some clues. While barnacles recruited in wave-exposed intertidal habitats outside of crevices typically die in the following winter because of intense ice scour (*MacPherson & Scrosati, 2008*; *Belt, Cole & Scrosati, 2009*), the extent of winter sea ice across the Gulf of St. Lawrence is decreasing (*Galbraith et al., 2024*). Therefore, warmer waters might not only favor barnacle recruitment in the spring but also a higher survival of the resulting adults as winter ice load decreases. Overall, barnacles might then retain higher abundances year-round in wave-exposed habitats on this coast, playing more importantly than before the traditional roles as prey, competitors, and facilitators known for milder temperate shores (*Dunkin & Hughes, 1984*; *Anderson, 1994*; *Harley, 2006*; *Menge et al., 2011*).

## ACKNOWLEDGEMENTS

We are grateful to Racine Rangel and an anonymous reviewer for constructive comments on an earlier version of this paper.

### Funding

This work was supported by the Natural Sciences and Engineering Research Council of Canada (NSERC) through a Discovery Grant (No. 311624) awarded to Ricardo A. Scrosati. The funders had no role in study design, data collection and analysis, decision to publish, or preparation of the manuscript.

### Grant Disclosures

The following grant information was disclosed by the authors:
The Natural Sciences and Engineering Research Council of Canada (NSERC): No. 311624.

### Competing Interests

The authors declare there are no competing interests.

### Author Contributions

- Ricardo A. Scrosati conceived and designed the experiments, performed the experiments, analyzed the data, prepared figures and/or tables, authored or reviewed drafts of the article, and approved the final draft.
- Julius A. Ellrich performed the experiments, analyzed the data, prepared figures and/or tables, authored or reviewed drafts of the article, and approved the final draft.

### Data Availability

  The dataset on barnacle recruitment is available at figshare: Scrosati, Ricardo A.; Ellrich, Julius A. (2024). Intertidal barnacle recruitment on the Gulf of St. Lawrence coast of Nova Scotia (2005-2016 and 2024). figshare. Dataset. https://doi.org/10.6084/m9.figshare.26332222.v1.

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
