# Peer review of "Massive barnacle recruitment on the Gulf of St. Lawrence coast of Nova Scotia (Canada) in 2024 linked to increased sea surface temperature"

_PeerJ, doi:10.7717/peerj.18208_

## Round 0.1 · original submission · Major Revisions

I have received two constructive reviews for your article. Both reviewers have useful suggestions for adding some context to the work in the introduction and discussion. Both reviewers also point to places to clarify some of the specific methods.

·

Basic reporting

Introduction
The authors begin in their abstract and within the introduction with a focus on thermal extremes and altered species performance, however, I think there are some missing pieces within the introduction on this aspect. Specifically, the intertidal can be a mosaic of thermal regimes which can influence recruitment success, predation, competition, performance that drive geographic variation in this habitat. I think it would benefit the authors to flesh out specifically how temperature can influence recruitment dynamics along coastlines. In the first paragraph the authors do touch on this (Lines 34-36) but then do not give any examples of how population abundance may increase with changing thermal environments. This can be particularly important when thinking about foundation species such as mussels (or barnacles) that are lost and replaced with non-foundational species leading to decreases in species richness and/or overall biodiversity.

Alternatively, the authors could restructure the first paragraph to be more about general population abundance changes across thermal environments and introduce the intertidal habitat later. This may be beneficial as paragraph 2 starts with field monitoring but then switches back to intertidal habitats and introducing barnacles. This reads quite disjointed.

Lines 71 – 74 are reading more as a result rather than an introduction as to why diet plays an important role in successful recruitment. I would suggest expanding on how feed plays a role in recruitment/how this is influenced by temperature as well.

Figures
Fig 1. I think it would be helpful to have a map with the site highlighted/pinpointed into this figure to aid the reader in the location of the transect and help the audience know the location if readers are not familiar with the study location.

Fig. 2 – Could the authors make the years before 2024 different shapes so the reader can have an idea of how much variation there is in the 2005-2016 dataset across time? Are all the years increasing over time? Or does it vary quite a bit? A legend may also allow readers to understand the figure more quickly without having to read the caption.

Experimental design

Methods
As the authors are heavily citing a previously published paper with the methods (Scrosati & Ellrich 2016) I found it hard to piece together their exact methods without having to look up the other paper. This makes reading this manuscript harder to follow and I suggest adding in more details of the methods without making the reader look up a previous paper. Such as – was any environmental data taken during the surveys? Did the surveys occur across multiple days?

I would also suggest the authors add a table summarizing the temperature data metrics (average +- SD for each year up till 2024) and the Chl-a concentrations. This could get moved to the results but there needs to be more information on the environmental variables in the dataset.

I would also like to know if the authors have any data on timing/duration of the ice scour? Did this vary across time and if so, did that influence the recruitment rate or density within the plots of barnacles measured in June? Is the barnacle recruitment increasing each year consistently across time?

There is also very little included on the statistical analyses conducted. What software/version was used to run these tests? Were all the assumptions met for the data - was it transformed in anyway?

Additionally, there is only the barnacle recruitment data listed here – what about the temperature and Chl-a data?

Validity of the findings

Results
Can the authors provide a table with the Welch tests parameters, and N of transects for each year? The authors state on lines 113-114 that the Chl-a was intermediate with a value of 8.3 mg m-3, however there is little context as what this value means in terms of barnacle recruitment and it appears there were years with even higher concentrations around 12.6 – were those years with higher barnacle recruitment? This information would be helpful to set up for readers earlier in the introduction.

Discussion
A piece of the discussion that I think is missing but important is how temperature may be influence faster recruitment/settlement rates of barnacles. The authors highlight how the larvae are planktonic for 5-6 weeks this may be changing if water temperature is increasing. So, it would be beneficial for the authors to increase the text in the discussion of how increasing temperature may be beneficial. Does this mean increased reproduction events during warmer seasons, increased egg production, or faster growing larvae?

Another discussion point that I think would be useful to include is other factors that can drive recruitment in barnacles including predation cues and adult cues. How can these interact with temperature to increase the patterns highlighted here by the authors?

The results/discussion of the food supply should be fleshed out a bit more. If the authors have the Chl-a data it would be beneficial to try and understand how much spatial or temporal variation there is in this metric and what that has looked like for the other years where barnacle recruitment was recorded. Is this consistent across other regions?

The authors bring up a really important point on lines 154-157 with loss of mussels. In this location has there been declines in mussel abundance? Are barnacle populations here also as sensitive to storms or temperature?

Additional comments

PeerJ – Manuscript #103775

Massive barnacle recruitment on the Gulf of St. Lawrence coast of Novia Scotia (Canada) with the warmer water of 2024

This manuscript aimed to highlight increases in barnacle (Semibalanus balanoides) recruitment/density from previous years (2005 to 2016) compared to those from 2024 after a larger increase in sea surface temperature (SST). The authors collected barnacle density data and utilized SST and Chl-a data. From this data collection they determined that the April 2024 SST which was 116% higher than the previous sampling period (4.1°C vs. 1.9°C) which led to an increase in barnacle recruitment which was 111% higher (607 to 1278 recruits dm^-2). They also found that in 2024, Chl-a exhibited an intermediate value compared with previous years and conclude that food did not play a role in barnacle recruitment in 2024. Overall, the authors found that increased SST in April 2024 played a vital role in barnacle recruitment on this coastline in Novia Scotia.

I do find this article interesting and thank the authors for their work and contributions. However, I have included some major and minor comments that I hope will help strengthen this manuscript and aid in their publishing this work. I do find that the current work may need some reworking of the introduction, results, and discussion before being published. Please find more detailed and minor comments below.

Minor comments
I would also suggest moving the beginning of paragraph 3 (starting on line 52) before discussing lines 50-51. Barnacles serve as indicators of nearshore pelagic conditions due to their pelagic larval periods.

Line 49 – Do the authors mean to cite Harley 2006?

Lines 88-89 were all barnacles counted here or specific sizes? The authors said recruits are all that are found here but what size class are they?

Line 93 – could the authors please include the link to these websites

Lines 133-136 – what does reproductive success mean here?

Lines 144 – 149 – could the authors try restructuring these sentences? They are a little vague and hard to follow.

Reviewer 2 ·

Basic reporting

This paper captures a pronounced increase in a common species of interest as a result of sea surface temperature increase. However, this manuscript would benefit from taking a broader scope that includes more of the basic ecology of the system and how it has/is changing as a result of climate change. The message of this paper would be strengthened by including more information about the other organisms present within the community. Further context about the intertidal zone itself would be helpful for readers that are not familiar with the system. Is this a highly invaded area? Who are the major players in the ecosystem? Is this increase in barnacle abundance the first signs that they are seeing regarding climate shifts, or is this one of many organisms that are responding to changing conditions? How have conditions changed and how are they expected to change in given years? How does this area in the Gulf of St. Lawrence compared to other areas regionally? Authors would benefit from citing additional papers that they themselves did not publish.

Experimental design

Authors clearly state that they are trying to compare an unusual recruitment episode of barnacles to 12 years of baseline data. Methods section focuses primarily on the 12 years of baseline data and mentions the collection procedure for the 2024 study at the end of the paragraph. I would suggest leading with the current study and then comparing to the previous studies.

Validity of the findings

The data presented are clear and straightforward. The discussion could be improved by tying these findings to the broader ecosystem and region. Are high recruitment events such as this a possibility? Who might this impact? How might competitors, predators, and the plankton community be impacted by large recruitment events? Have there been other large recruitment events or failures as a result of high sea surface temperatures?

Additional comments

The authors state that the majority of recruitment occurs in May, yet the 2024 sampling took place in July, while other sampling occurred in June. Might they be understating levels of recruitment in 2024? Are there other studies looking at early rates of mortality from overgrowth, disease or predation? Do the authors have any thoughts about whether this unusual recruitment event was a result of increased larval production, larval survival, or recruitment success? Did anyone measure barnacle abundance in the region between 2016 and 2024? I feel that this manuscript would benefit from providing an argument about what this increase in barnacle settlement might mean for the area. Is this one isolated instance of increased abundance as a result of warm water temperatures, or is this one of many such events?

Annotated reviews are not available for download in order to protect the identity of reviewers who chose to remain anonymous.

---

## Round 0.2 · Minor Revisions

I appreciate the revised document and addressing reviewer concerns. There are a couple of areas where I felt the reviewers brought up good points that were brushed aside. Please address the following: adding a map to figure 1 and adding more details on the methods (the questions by reviewer 1: was any
environmental data taken during the surveys? Did the surveys occur across multiple days? are both reasonable. The reviewer is not asking to repeat ALL details of the previous paper)

---

## Round 0.3 · accepted · Accept

Thank you for addressing reviewer concerns. The manuscript is ready for publication.